# Chronic Kidney Disease: Strategies to Retard Progression

**DOI:** 10.3390/ijms221810084

**Published:** 2021-09-18

**Authors:** Ming-Tso Yan, Chia-Ter Chao, Shih-Hua Lin

**Affiliations:** 1Department of Medicine, Division of Nephrology, Cathay General Hospital, School of Medicine, Fu-Jen Catholic University, Taipei 106, Taiwan; qqhaibear@yahoo.com.tw; 2National Defense Medical Center, Graduate Institute of Medical Sciences, Taipei 114, Taiwan; 3Department of Internal Medicine, Nephrology Division, National Taiwan University Hospital, Taipei 104, Taiwan; b88401084@gmail.com; 4Graduate Institute of Toxicology, National Taiwan University College of Medicine, Taipei 104, Taiwan; 5Department of Internal Medicine, Nephrology Division, National Taiwan University College of Medicine, Taipei 104, Taiwan; 6Department of Internal Medicine, Nephrology Division, National Defense Medical Center, Taipei 104, Taiwan

**Keywords:** acute kidney injury, chronic kidney disease, renal progression, therapy for renal failure

## Abstract

Chronic kidney disease (CKD), defined as the presence of irreversible structural or functional kidney damages, increases the risk of poor outcomes due to its association with multiple complications, including altered mineral metabolism, anemia, metabolic acidosis, and increased cardiovascular events. The mainstay of treatments for CKD lies in the prevention of the development and progression of CKD as well as its complications. Due to the heterogeneous origins and the uncertainty in the pathogenesis of CKD, efficacious therapies for CKD remain challenging. In this review, we focus on the following four themes: first, a summary of the known factors that contribute to CKD development and progression, with an emphasis on avoiding acute kidney injury (AKI); second, an etiology-based treatment strategy for retarding CKD, including the approaches for the common and under-recognized ones; and third, the recommended approaches for ameliorating CKD complications, and the final section discusses the novel agents for counteracting CKD progression.

## 1. Introduction

Chronic kidney disease (CKD) is defined as a progressive and irreversible loss of renal function evidenced by an estimated glomerular filtration rate (eGFR) of <60 mL/min per 1.73m^2^, the persistent presence of manifestations that are suggestive of kidney damage (proteinuria, active urine sediments, histological damages, structural abnormalities or a history of kidney transplantation), or both, lasting for more than 3 months [1]. CKD has long been a worldwide public health concern and constitutes a heavy healthcare and economic burden, as a reduced GFR is widely known to increase the risk of cardiovascular events, hospitalization, cognitive dysfunction, and overall mortality [2]. The prevalence of CKD varies according to geographic areas, mostly ranging from 10 to 20%, but rises gradually, particularly in developed countries [3,4,5]. This trend can be partially attributed to the expanding aging population globally [6]. In addition, the increased prevalence of risk factors such as diabetes mellitus (DM), hypertension, and obesity in patients with CKD is also notable [7,8].

Having a diagnosis of CKD means that an individual’s renal function has entered into a “point of no return,” indicating that the deterioration of renal function over time is inevitable and frequently irreversible. However, there can still be different patterns of renal function decline in patients with CKD. These patterns can be classified intuitively into very fast, fast, moderate, or slow, depending on the threshold required for defining the renal function decline rate, using mL/min/year or mL/min/month (Figure 1A). These differences in patterns largely reflect the heterogeneity of CKD origins and the subsequent pathologies, adjunct comorbidities, interventions that patients receive, and other harsh environmental exposures [9]. From this perspective, finding how to retard renal progression in patients with CKD is still fraught with challenges. In this review, we will focus on established and potential management strategies that aim to slow down the deterioration of renal function and treat the relevant complications in patients with CKD.

## 2. Risks of CKD Progression

Estimated GFR trajectories are highly variable in CKD. This phenomenon implies that a wide array of heterogeneous risk factors may contribute to CKD progression and that these risk factors may be potential therapeutic targets if we wish to achieve renoprotection. Socioeconomic factors and lifestyle factors (e.g., diet, sleep deprivation, smoking, and lack of exercise) are well-known risk factors that are associated with CKD progression. Systemic and metabolic disorders, including DM, hypertension, gout, and cardiovascular diseases, can also precipitate the development of CKD and aggravate eGFR decline (Figure 2) [10]. Recently, atrial fibrillation (AF) has also been shown to be a contributor to rapid eGFR decline, and the CHA_2_DS_2_-VASc score, a stroke-risk stratification model for patients with AF, can predict renal progression [11].

The intrinsic factors that are related to kidneys per se also play an important role in influencing renal function decline, such as GFR, proteinuria, glomerulopathy, interstitial lesions, and renal outlet obstruction (obstructive nephropathy). Each glomerulopathy distinctly influences the pace of renal function deterioration; for example, focal segmental glomerulosclerosis (FSGS) is more likely to result in a faster GFR decline, while the rate is lower for patients with IgA nephropathy (IgAN), membranous nephropathy (MN), and diabetic kidney disease (DKD) [12]. Polymorphisms in the genes involved in pathways such as those associated with inflammatory reactions (e.g., tumor necrosis factor (TNF)-α and interleukin (IL)-4), fibrosis (e.g., TGFB1), phase-II metabolism (e.g., GSTP1 and GSTO1), CKD worsening (e.g., UMOD) and the renin–angiotensin–aldosterone system (RAAS) (e.g., AGT and RENBP) have been suggested to affect the progression rate of patients with CKD [13].

## 3. Epithelial–Mesenchymal Transition (EMT)

Renal fibrosis, including nephrosclerosis and tubulointerstitial fibrosis, constitutes the final common pathway of renal injuries, regardless of etiologies. EMT is the major mechanism promoting renal fibrosis, and myofibroblasts are the main cell type that produces the extracellular matrix [14]. The origin of myofibroblasts in the kidney remains uncertain, but several candidates have been suggested, including resident fibroblasts, bone marrow-derived fibroblasts, or transition from pericytes or endothelial cells [15]. Recent research has revealed that EMT is fairly uncommon, as fibroblasts derived from EMT are rarely found in renal interstitium. A novel concept of partial EMT, indicating that tubular epithelial cells gain mesenchymal characteristics but retain their attachment to the basement membrane, may explain the pathogenic role of renal tubular epithelia in renal fibrosis [16].

After acute kidney injury (AKI) attacks, the c-Jun NH2-terminal kinase (JNK) signal is activated in tubular epithelial cells to enhance the expression of typical mesenchymal markers (e.g., e-cadherin, α-smooth muscle actin) and to upregulate profibrogenic factors (mainly transforming growth factor (TGF)-β and connective tissue growth factor (CTGF)) [17]. The persistent activation of the TGF-β pathway leading to an increased expression of SNAI1 and TWIST1 further promotes G2/M arrest. Cell cycle arrest in the G2/M phase in injured tubular cells, through JNK activation, amplifies the profibrogenic factors, such as TGF-β and CTGF, constituting a vicious cycle culminating in the fibrosis progression [18]. Fatty acid oxidation (FAO) is the main energy source of the proximal tubule (PCT). SMAD3 activated by TGF-β will suppress PPARGC1a expression to cause dysregulated FAO with lipid accumulation in the PCT, one of the characteristic features of EMT [19]. The PCT cells with lipid accumulation will enhance inflammation, innate immunity, and apoptosis to worsen renal fibrosis. Tubular cells with partial EMT can also activate fibroblasts and recruit inflammatory cells via the secretome composed of growth factors, chemokines, and cytokines, subsequently aggravating fibrosis [20]. Blocking EMT with approaches targeting the cell cycle or the inhibition of SNAI1 or TWIST1 expressions has been found to repress inflammation and fibrosis, pointing to the fact that EMT may be a good target mechanism to reverse renal fibrosis [15,18]. Bone morphogenetic protein-7 (BMP-7) can reverse EMT by counteracting the TGF-β/SMAD2/3 pathway and serves as another potential therapeutic target for improving renal injury [21]. However, the results in clinical studies of BMP-7 analogs involving patients with CKD are heterogeneous, suggesting a complex interaction between BMP-7 and other EMT-related pathways, as well as the necessity of determining the optimal serum BMP-7 concentration [22].

Epigenetic modifications, including DNA methylation and histone modification, also participate heavily in the regulation of partial EMT. The inhibition of DNA methylation was reported to ameliorate renal fibrosis. For example, low-dose hydralazine causing the de-methylation of the NASAL1 promotor and 5′-azacytidine, resulting in an inhibition of DNA methyltransferase 1 (DNMT1) [23,24,25]. Furthermore, agents targeting histone modification also confer renal benefits in CKD or AKI-to-CKD transition through the inhibition of histone methyltransferase (e.g., enhancer of zeste homolog 2) or the inhibition of histone deacetylases by directly inhibiting deacetylase (e.g., valproic acid) or indirectly interfering with histone modification readers (e.g., bromodomain and extra-terminal (BET) protein inhibitors) [26,27,28]. Epigenetics may be a novel therapeutic target for renal diseases.

## 4. Avoidance of Acute Kidney Injury (AKI) in Patients with CKD

AKI is associated with significant morbidity and mortality, including an increase in adverse renal outcomes. Cumulative evidence suggests that AKI is never self-limited, as it serves as a gateway to subsequent AKI episodes and, potentially, incident CKD, regardless of whether patients show recovery from AKI episodes or not [29]. Moreover, a single episode of severe AKI superimposed on patients with pre-existing CKD can cause further renal deterioration to end-stage kidney disease (ESKD) rapidly at a non-linear pace (Figure 1B). The risk factors associated with the AKI-related acceleration of renal progression were identified previously, and include older age, delayed renal function recovery from AKI, severe AKI episodes, the presence of proteinuria, and comorbidities such as DM, hypertension, and heart failure [30,31,32,33,34,35,36,37,38,39,40,41,42,43,44,45,46,47,48,49,50,51,52,53,54,55,56,57,58,59,60,61,62,63,64,65,66,67,68,69,70,71,72,73,74,75,76,77,78,79,80,81,82,83,84,85,86,87,88,89,90,91,92,93,94,95,96,97,98,99,100,101,102,103,104,105,106,107,108,109,110,111,112,113,114,115,116,117,118,119,120,121,122,123,124,125,126,127,128,129,130,131,132]. Long-term renal sequelae become very serious if patients have acute tubular necrosis (ATN) or ischemic AKI compared with those who have other forms of AKI, suggesting that the etiology of the AKI may also be a very important risk factor [33].

Biomarkers such as kidney injury molecule (KIM)-1 and neutrophil gelatinase-associated lipocalin (NGAL) are able to detect AKI earlier than conventional indicators [34,35]. Moreover, a combination of biomarkers such as insulin-like growth factor-binding protein 7 (IGFBP7) and tissue inhibitor of metalloproteinases-2 (TIMP-2) was shown to successfully predict the development of AKI during the 12 h following blood tests and can guide the selection of interventions to prevent AKI [36,37].

The mechanisms that are responsible for the transition from AKI to CKD remain under active investigation, and a maladaptive repair response with partial epithelial–mesenchymal transition (EMT), especially at the proximal tubule (PCT), has an important role. Cell cycle arrest in the G2/M phase, the dysregulated regeneration of injured PCT due to mitochondrial dysfunction, and the aberrant activation of developmental pathways (such as the Wnt, Hedgehog, and Notch pathways) also contribute to AKI-to-CKD transition. Phenotypic changes of fibroblasts to myofibroblasts or the formation of tertiary lymphoid tissue, as well as defective switches of recruited T cells and M1 macrophages to regulatory T cells and M2 macrophages, respectively, will perpetuate inflammation and fibrosis, leading to capillary rarefaction, hypoxia, and tubule cell damage, constituting a vicious cycle [38,39,40]. Furthermore, the persistent expression of either transforming growth factor (TGF)-β or kidney injury molecule-1 (KIM-1) has also been held responsible [41,42]. Notably, several pathogenic processes contributing to AKI-to-CKD transition are also physiological repair processes within kidneys. Additional insults, including high salt and high protein intake and nephrotoxic agents, during or after AKI episodes, as well as the underlying diseased kidney (e.g., diabetic nephropathy), may turn physiological renal responses into disordered regeneration.

### Identification of Factors Causing AKI

The fundamental management of AKI in patients with CKD involves preventing the occurrence of AKI, and the success of this approach largely depends on the identification of the etiologies contributing to AKI (Table 1). Furthermore, treating the underlying etiologies of AKI, the optimization of volume and hemodynamic status, the withdrawal of nephrotoxic agents, the adjustment of medication doses according to renal function, adopting a conservative (<180 mg/dL), rather than an intensive glycemic control goal, and maintaining higher mean arterial pressure in patients with underlying hypertension can be crucial strategies for AKI management [43,44]. Drug regimens should be reviewed carefully to prevent drug–drug interactions. For example, adding piperacillin/tazobactam should be avoided, as it potentiates the nephrotoxicity of vancomycin. Several agents and strategies have been tested for treating AKI, including recombinant alkaline phosphatase and L-carnitine for sepsis-related AKI, as well as p53-targeted small interfering RNA (siRNA) and remote ischemic preconditioning for surgery-associated AKI [45,46].

## 5. An Etiology-Based Treatment Strategy for CKD

As mentioned above, the different etiologies of CKD themselves have various impacts on renal progression. As an etiology-based treatment strategy for CKD has not been well addressed, we touch on this management issue for both the common and less-appreciated causes of CKD.

### 5.1. Glomerulopathy

Glomerulopathy is a heterogeneous group of diseases and accounts for a significant number of CKDs. It occurs more commonly in young people with non-specific presentations. Although novel diagnostic tools are under investigation, renal biopsy is still the gold standard for achieving a definite diagnosis. A slow deterioration of renal function occurs in a proportion of patients. The risk factors for a faster GFR reduction include obesity, smoking, hypertension, significant proteinuria (usually >1 g/day), CKD at the diagnosis of glomerulopathy, and pathologically chronic renal lesions (glomerulosclerosis, tubular atrophy, and interstitial fibrosis) [47]. Genetic factors contribute to a rapid GFR loss, such as *APOL1* in those with FSGS [48]. Fabry disease is a frequently ignored X-link inherited disease caused by a pathogenic mutation involving *GLA*-encoding lysosome enzyme α-galactosidase A [49]. The deficient enzyme function results in the intracellular accumulation of globotriaosylceramide, which impairs cell metabolism. Apart from neurological and cardiovascular associations, the kidneys may also be affected by presentations of proteinuria and renal failure. Enzyme replacement therapy is the cornerstone of renal progression reduction or prevention. A brief summary of the preferred treatment for different underlying glomerulonephritis is shown in Table 2.

### 5.2. DM-Related CKD

DM is the most common etiology of CKD and ESKD worldwide [50]. DKD usually occurs in patients with poor glycemic control, but also arises in 30–40% of patients with intensive glycemic control, suggesting a complex and multifactorial pathogenesis of DKD [51,52,53]. Clinical manifestations of DKD include impaired renal function with proteinuria. Several risk factors have been discovered, including early onset of DM, hypertension, ethnicity, obesity, the severity of proteinuria, and smoking [54,55,56,57]. In addition to adequate glycemic control, RAAS blockade is the centerpiece of DKD management. Pentoxiphylline can delay the initiation of dialysis and exert a significant antiproteinuric effect in DM patients already receiving RAAS blockades [57]. Although most statins exhibit minimal renoprotective effects [58,59,60], fenofibrate, a peroxisome proliferator-activated receptor (PPAR) α-agonist, has been shown to have an antiproteinuric effect in DKD [61]. Thiazolidinediones, a PPARγ agonist, can be beneficial for DKD, with salt and fluid retention side effects [62,63] that should be considered.

Sodium-glucose cotransporter-2 (SGLT2) inhibitors are the latest anti-hyperglycemic agent capable of decreasing blood glucose effectively by blocking glucose reabsorption in the PCT. In addition to cardiovascular benefits, SGLT2 inhibitors also postpone renal deterioration and reduce the severity of proteinuria in patients with DM (Figure 3) [64,65,66]. It is proposed that SGLT2 inhibitors protect the kidney by enhancing glycemic control, improving cardiovascular function, and decreasing body weight, as well as restoring intra and extrarenal hemodynamics, including lowering blood pressure (BP), promoting natriuresis, and re-activating tubuloglomerular feedback [67]. Structurally, SGLT2 inhibitors also are known to reverse PCT hypertrophy, which is induced by insulin resistance and hyperglycemia-related increasing sodium and glucose reabsorption through SGLT2 [68]. With the reversal of PCT hypertrophy and decreased Na+-glucose reabsorption, the kidney will be protected due to reduced energy demand and subsequently less oxidative stress, inflammation, fibrosis, and growth factor expression. Furthermore, AMPK/SIRT1 signaling, which is suppressed by hyperglycemia, could be re-activated by SGLT2 inhibitors to promote anti-inflammatory hypoxia-inducible factor (HIF)-2α and to suppress the expression of pro-inflammatory HIF-1α [69,70,71]. Hyperglycemia increases the production of reactive oxygen species (ROS), diacylglycerol (DAG), and advanced glycation end products (AGEs), all contributing to the impaired autophagic clearance of SNAI1 and activated p21 and p27. Accumulated SNAI1 and the activation of p21 and p27 result in G2/M cell cycle arrest, the hallmark of EMT, and maladaptive renal tubular regeneration. The use of SGLT2 inhibitors can restore normal autophagic clearance and inhibit pathways due to the products of hyperglycemia, attenuating cell cycle arrest-related kidney damage. Notably, it is found that the renoprotective effect may be extended to non-diabetic patients with kidney diseases [72].

O-GlcNAcylation is a post-translational modification of proteins and can regulate various physiological or pathological processes. In DM, O-GlcNAcylation has a role in insulin resistance and can mediate glucose toxicity, and subsequently, DM complications, including DKD. Recent research has demonstrated that the O-GlcNAcylation of cellular proteins such as ICln impairs cell volume regulation in diverse cell types, a common phenomenon in DM [73]. A hypertrophic PCT is a typical feature of DKD as increased filtrated glucose promotes the reabsorption of glucose and sodium in PCT. O-GlcNAcylation may result in the cell death of PCT and contribute to the development or progression of DKD by impairing the volume regulation, which could be reversed by reducing the O-GlcNAcylation of ICln [74]. Moreover, O-GlcNAcylation in PCT was found to correlate with fatty acid oxidation [75]. These findings suggest that the manipulation of O-GlcNAcylation may be a potential target for the treatment DKD [76].

## 6. Hypertension-Related CKD

BP lowering is one of the most important managing strategies for hypertensive patients with CKD [77]. The previous consensus suggested that a systolic BP (SBP) goal of <140 mmHg in patients with CKD and <130 mmHg in patients with CKD and proteinuria [78] would be reasonable. In proteinuric CKD patients, intensive BP control is associated with a lower occurrence of serum creatinine doubling or ESKD. Of note, Kidney Disease Improving Global Outcomes (KDIGO) updated the treatment target of SBP control to <120 mmHg in patients with hypertensive CKD [79].

Besides lifestyle modifications, including salt restriction, smoking cessation, weight loss, and adequate exercise, pharmacotherapy can be beneficial with the use of renoprotective agents, including RAAS blockades and carvedilol [80,81,82]. Mineralocorticoid receptor antagonists (MRAs), such as spironolactone, are able to reduce proteinuria and BP, but are limited in the treatment of CKD due to their association with GFR decline and hyperkalemia [83,84,85]. Novel nonsteroidal MRAs, such as esaxerenone and finerenone have been developed with better anti-fibrotic and anti-inflammatory effects (Figure 4) [86,87]. Compared with steroidal MRAs, both finerenone and esaxerenone exhibit similar proteinuria-lowering effects in patients with CKD, and a lower incidence of hyperkalemia [88,89,90]. Moreover, a combination of one RAAS blockade with nonsteroidal MRAs showed a similar AKI occurrence with finerenone but steadily decreased GFR in esaxerenone trials [91,92]. Whether the combination of nonsteroidal MRAs with other RAAS blockades will confer further renoprotection merits further investigation.

### 6.1. Heart Failure (HF)-Related CKD

In addition to traditional medications to control HF, sacubitril, an angiotensin receptor–neprilysin inhibitor (ARNI), is currently recommended for patients with HF due to its significant benefits in reducing cardiovascular mortality and hospitalization [93]. In view of the frequent occurrence of impaired renal function in patients with heart failure, the potential renal advantages of sacubitril/valsartan are being investigated. Sacubitril/valsartan was recently found to help reserve renal function and to reduce the severity of proteinuria in HF patients with a reduced ejection fraction [94].

### 6.2. Nephrolithiasis and Urothithiasis

Nephrolithiasis significantly increases the risk of incident CKD and accounts for about 2–3% of cases with ESKD [95]. Stone formers have a lower estimated GFR compared with those without renal stones [96]. This phenomenon likely results from the fact that nephrolithiasis shares many risk factors with CKD, including nephrotoxic analgesic use for pain control during obstructive uropathy, a decreased water intake leading to volume depletion, an increased dietary protein intake, recurrent sepsis, urinary tract structural abnormalities, and exposure to contrast media for imaging purposes [97,98]. Notably, both operations and shock wave lithotomy induce renal parenchymal injury, inflammation, and fibrosis. Furthermore, different stone types correlate with different CKD risk. For example, cystine stones carry the highest risk of CKD progression, followed by uric acid and struvite stones [99]. Given the higher rate of recurrence, nephrolithiasis and urothithiasis in CKD should be well managed.

Pathways leading to CKD may be stone-specific. Brushite stones frequently generate plugs at the ducts of Bellini and cause duct obstruction. The deposition of uric acid crystals may lead to inflammation and fibrosis. Chronic pyelonephritis, high urine pH due to urease action, and staghorn stones in the struvite stone formers may cause papillary necrosis and renal parenchymal injury [100]. Uncommon hereditary diseases can also present with nephrolithiasis, such as primary hyperoxaluria, cystinuria, Dent’s disease, and adenine phosphoribosyltransferase (APRT) deficiency [101,102]. The prevention of stone formation may be a key step to improve outcomes (Figure 5). Patients with uric acid stones have also been found to have better renal outcomes when they receive xanthine oxidase inhibitors, especially febuxostat and maintain alkaline urine [103]. Stone-specific agents can be utilized. Lumasiran, an RNAi therapy for type 1 primary hyperoxaluria, was recently approved for clinical use [104].

### 6.3. Autosomal Dominant Polycystic Kidney Disease (ADPKD)

ADPKD is the most common genetic cause of ESKD [105,106] and the majority of cases carry mutations in either *PKD1* or *PKD2*. Recently, *GANAB* encoding glucosidase II subunit α was identified as another pathogenic gene [107]. Clinical manifestations of ADPKD include renal (flank pain, hematuria, calculi, urinary tract infection, polyuria, nocturia, and hypertension) and extrarenal (cerebral aneurysm, cysts in the liver or other organs, and valvular heart disease) presentations [108]. Multiple signaling pathways have been suggested to cause metabolic disturbance during the course of ADPKD, especially the cAMP pathway, which serves as the central player in cystogenesis [109] Transplantation is the best therapeutic strategy since treatments such as angiotensin-converting enzyme inhibitor (ACEi) or angiotensin receptor blocker (ARB) provide a limited effect on GFR decline [110]. However, encouraging results suggest that the use of the V2 receptor antagonist tolvaptan in patients aged <55 years with an estimated GFR >25 mL/min/1.73 m^2^ can delay the worsening of kidney function and decrease the volume in a dose-dependent manner with adequate safety and tolerance [111,112].

### 6.4. Autosomal Dominant Tubulointerstitial Kidney Disease (ADTKD)

Autosomal dominant tubulointerstitial kidney disease (ADTKD) is characterized by tubular damage and interstitial fibrosis with intact glomeruli and positive family history [113]. It inevitably causes CKD progression to ESKD but is usually un-recognized even though it accounts for about 5% of monogenic disorders resulting in ESKD [114]. The introduction of genetic tests improves the sensitivity of the diagnosis. The causative mutations mainly involve five genes, including *UMOD*, *MUC1*, *REN*, *HNF1B*, and *SEC61A1*. Although there is no specific therapy available for ADTKD currently, a low-salt diet is not suggested and diuretics should be used with caution to prevent the aggravation of salt and volume depletion as well as hyperuricemia and gout [115]

### 6.5. Patients with Graft Kidney

Compared with dialysis, patients receiving kidney transplantation have better outcomes and quality of life. After induction therapy with T cell depleting agents, maintenance immunosuppressants targeting three signals for T cell activation and proliferation should be continued to avoid rejection [116]. Glucocorticoids are commonly used for the induction and maintenance of immunosuppression by inhibiting nuclear factor κB (NF-κB) and its downstream cytokine expression. Early withdrawal can be considered in patients without direct immune-mediated renal diseases and with low immunologic risk [117]. Antimetabolites are also commonly used, including azathioprine, mycophenolate/mizoribine, and its active metabolite [118].

Calcineurin inhibitors (CNIs) are the major medications blocking signal 1 via binding to the FK506 binding protein (FKBP). Although CNIs improve graft outcomes significantly, 12 h trough or 2 h peak (for neoral only) serum levels should be measured to prevent excessive nephrotoxicity [119]. The complex drug interactions of CNIs with those capable of inducing or inhibiting cytochrome P450 need more attention. mTOR inhibitors (mTORi) also bind to FKBP but inhibit mTOR from blocking signal 3 [120]. Proteinuria should be measured during the use of mTORi to monitor side effects such as de novo proteinuria [121]. Belatacept binding to CD80/CD86 on antigen-presenting cells can interrupt the interaction with CD28 on T cells to block signal 2 [122].

Triple therapy is usually maintained in patients with higher immunologic risk, such as those with underlying glomerulonephritis, receiving re-transplantation, and those with high panel-reactive antibody titers [123]. With a lower estimated GFR after transplantation, a mTORi-based regimen is considered more often than a CNI-based regimen [124]. mTORi can replace CNIs or antimetabolites with similar allograft survival, better post-transplant renal function reserve, but a decreased occurrence of non-melanoma skin cancers [125]. Except for de novo proteinuria, mTORi may contribute to hyperlipidemia, bone marrow suppression, and infection. Other than pharmacotherapy, the frequent monitoring of kidney graft function is crucial. While serum creatinine increases, it is important to identify reversible factors such as sepsis, volume depletion, and drug toxicity. BK virus infection also needs to be excluded. Ultrasound is helpful in evaluating structural problems and vascular inflows. Kidney allograft biopsy may be indicated if recurrent/de novo kidney diseases or rejection are suspected, or incident proteinuria occurs.

## 7. Management of CKD Complications

The clinical manifestations of CKD are different according to the etiologies, staging, and comorbidities. Kidneys not only handle solute excretion and water balance but also maintain endocrine homeostasis. As CKD progresses, uremic toxins will accumulate. The appropriate management of CKD complications may also attenuate the renal progression rate.

### 7.1. Metabolic Acidosis

With the progressive loss of renal function, the declined renal capacity of acid excretion and ammonia synthesis, along with increased non-volatile acid production, leads to metabolic acidosis. The prevalence of metabolic acidosis was shown to increase linearly with GFR decline [126]. Metabolic acidosis can contribute to renal progression, and using alkali therapy was shown to slow renal progression [127]. The mechanism of metabolic acidosis-induced kidney progression is multi-faceted (Figure 6). Enhanced ammonia production in surviving nephrons secondary to metabolic acidosis in surviving nephrons may result in complement activation with tubulointerstitial damages [128,129]. Increased endothelin production related to metabolic acidosis is found to reduce GFR and tubulointerstitial injury [130]. Beyond renal adverse effects, metabolic acidosis negatively influences cardiovascular outcomes by aggravating inflammatory responses, increasing aldosterone secretion, enhancing endothelin synthesis, and impairing endothelial function, as well as reducing Na^+^/K^+^-ATPase activity and subsequently impaired heart contractility [131,132]. Metabolic acidosis is also associated with impaired bone mineralization, insulin resistance, and higher all-cause mortality [133,134].

Alkali therapy can be initiated early to achieve a serum HCO_3_^−^ level between 22 and 26 mEq/L [135], since a HCO_3_^−^ concentration >26 mEq/L is associated with higher mortality and cardiac events [136]. NaHCO_3_^−^ and Na citrate are two commonly used alkali supplements; the former is cheaper but may result in bloating, while the latter is costly and enhances gastrointestinal absorption of aluminum. The concern about sodium/volume retention and hypertension being exacerbated by alkali therapy may be relieved with diuretics. Fruits and vegetables have also been shown to significantly increase serum HCO_3_^−^ levels and to help preserve GFR in patients with stage 3 to 4 CKD without producing hyperkalemia, despite their lower effectiveness compared to medications [137].

### 7.2. Low Protein Diet with Ketoanalogues

A high protein intake causes hyperfiltration and increased intraglomerular pressure, resulting in the onset or progression of CKD. Therefore, dietary protein restriction has long been thought as the mainstay of nutritional therapies for CKD. In general, dietary protein of 0.55–0.6 g/kg/day is suggested for stage 3 to 5 CKD patients without DM, and 0.6–0.8 g/kg/day for those with DM. Compared with animal proteins, plant proteins have less influence on glomerular hemodynamics and a lower net acid production. Phosphate in plant origin proteins has relatively low bioavailability and results in a smaller phosphate accumulation [138]. Clinically, plant proteins are associated with a reduced rate of GFR decline and better outcomes and may be preferred as the major protein source in patients with CKD [139,140].

As a very low protein diet (0.3–0.4 g/kg/day protein) was reported to reduce the risks of incident dialysis compared with a low or normal protein diet, the means of preventing protein-energy wasting during a restriction of dietary protein becomes a critical issue [141]. Ketoanalogues are the precursors of essential amino acids, and their use with concomitant dietary protein restriction has been found to significantly delay the progression of CKD and to reduce the risks of dialysis initiation in patients with (estimated GFR >18 mL/min/1.73 m^2^) or without advanced CKD [142].

Regular physical exercise is recommended for patients regardless of CKD stage and may benefit kidney outcomes. However, there is still doubt about the relationship between the restriction of dietary protein and uremic sarcopenia in CKD patients. Indeed, regular physical activity during a low protein diet or a very low protein diet with ketoanalogues will not cause net protein catabolism but will help with an improvement in muscle strength, inflammation, and nutrition status if the energy supply is adequate [143,144]. Therefore, an increased intake of protein-free nutrition provides adequate energy needs without the trade-off of a high phosphate intake. Notably, the use of ketoanalogues can effectively prevent the occurrence of protein-energy wasting and may be considered as an important part of CKD nutritional therapy.

### 7.3. Anemia

Anemia is a frequent complication of CKD and is associated with a poor prognosis. The correction of anemia in CKD may enhance the quality of life for patients and reduce their renal function deterioration. Presently, mechanisms that are responsible for the development of renal anemia are only partially known [145]. Insufficient erythropoietin (EPO) production has been suggested as the major cause of renal anemia. Although the introduction of erythropoiesis-stimulating agents (ESAs) reduces transfusion-related complications and improves the symptoms of anemia [146], no significant improvement in anemia-related mortality and morbidity is observed, which is probably related to their adverse effects, such as worsening hypertension, malignancy progression, higher cardiovascular complications, and thrombosis [147]. These findings suggest that EPO deficiency is not the sole cause of renal anemia.

Hepcidin, a 25-amino acid peptide encoded by *HAMP*, serves as a key molecule in iron homeostasis by reducing gastrointestinal iron absorption and compromising iron distribution [148]. Inflammatory cytokines, such as interleukin-6, enhance hepcidin production, while EPO increases erythroferrone expression to suppress hepcidin production [149,150]. Hepcidin also plays a central role in iron regulation in patients with CKD; either chronic inflammation or EPO deficiency related to CKD enhances hepcidin production, resulting in impaired intestinal iron absorption and the mobilization of iron from its storage site [151,152].

Recently, hypoxia-inducible factor (HIF) stabilizers, such as roxadustat and vadadustat, have been developed for the treatment of renal anemia [153]. HIF, unlike ESAs, can bind to specific sequences called hypoxia response elements (HREs) to increase the production of endogenous EPO and to improve iron utility under hypoxia. With normoxia, the hydroxylation of HIF by prolyl hydroxylase domain (PHD) enzymes will cause the degradation of HIF for the purpose of regulating HIF activities [154]. HIF stabilizers have been shown to be non-inferiority to ESAs, especially in non-dialysis CKD, with regards to anemia-treatment efficacy [155,156]. However, their pleiotropic effects and long-term complications still require further investigation.

### 7.4. Hyperkalemia

Hyperkalemia is a potentially life-threatening condition and correlates with a rapid decline in renal function in patients with CKD. To educate patients with CKD about the restriction of dietary potassium (K^+^) is the cornerstone of management. Although cationic exchange resins, including sodium or calcium polystyrene sulfonate have been widely used in patients with CKD, the associated sodium load, gastrointestinal irritation, inadequate selectivity for certain cations, and their limited therapeutic efficacy reduce the utility of these resins. Recently, novel agents, including sodium zirconium cyclosilicate (ZS-9) and patiromer have been shown to have a higher selectivity for potassium with milder GI adverse effects. ZS-9 may play an important role in the management of acute hyperkalemia, due to its rapid onset of action (within 1 h), while patiromer shows a persistent K^+^-lowering effect lasting for 48 h [157,158,159]. Importantly, the high efficacy of both ZS-9 and patiromer to lower K^+^ may improve the prognosis of patients with CKD by reducing the probability of discontinuing RAAS blockades resulting in better heart and renal outcomes.

## 8. Specific Issues of CKD

### 8.1. Novel Therapeutic Approaches

The limited effects of the current therapy for CKD drives the need for the development of novel therapeutic agents. Regardless of the etiologies, several processes, including fibrosis, inflammation, and oxidative stress, as well as impaired cell regeneration, are ongoing in the kidney. Some novel agents targeting these pathways were investigated in preclinical studies and clinical trials. For example, atrasentan targeting the endothelin-1 receptor ETA, successfully delays the progression of DKD in clinical trials [160]. Given the critical role of transforming growth factor (TGF)-β in not only fibrosis but also inflammation, oxidation, and apoptosis, pirfenidone (with the ability to reduce TGF-β production) was shown to retard the progression of estimated GFR in FSGS clinical trials [161]. MicroRNAs can regulate numerous biological processes by repressing translationally or mediating the degradation of mRNA. MiR-21 can be induced by TGF-β in kidneys and is associated with fibrosis and podocyte injury. Lademirsen, as a blocker of miR-21, was found to prevent the progression of Alport syndrome in preclinical studies [162]. Bromodomain and extraterminal (BET) proteins are epigenetic regulators involved in cell proliferation, differentiation, and inflammation. A BET protein inhibitor, apabetalone, showed favorable renal outcomes in patients with coronary artery disease and DKD [163,164]. Furthermore, clinical trials using inhibitors of Nrf2 and p53 are ongoing [165].

Considering the complexity of CKD pathogenesis involving different cell types and changes of multiple signal pathways, multi-target drugs (MTD) emerge as a useful tool; for example, soluble epoxide hydrolase (SHE)-based PTUPB (with cyclo-oxygenase 2 (COX2) inhibitor), PB394 (with PPARγ agonist), and DM509 (with farnesoid X receptor agonist) to alleviate fibrosis, inflammatory response, and oxidative stress from CKD related to DM, hypertension, hyperlipidemia, or other etiologies [166,167,168].

### 8.2. Mesenchymal Stem Cells and Their Conditioned Media

Mesenchymal stem cells (MSCs) characterized by the ability to self-renew and differentiate to different cells can be isolated from different tissues such as adipose tissue and bone marrow [169]. Due to the low immunogenicity of MSCs, their transplantation is a safe therapy and has been used in the treatment of numerous diseases, including kidney diseases. Transplanted MSCs can migrate into injured tissue, so-called MSC homing, and exert their immune modulation, anti-apoptosis, anti-inflammation, and anti-oxidation abilities to enhance tissue repair in the manner of direct cell–cell interaction or paracrine [170]. Moreover, extracellular vesicles (EVs) secreted by MSCs and the conditioned medium (CM) of MSCs are both shown to help heal injured tissues [171]. In diverse models of AKI and CKD, the systemic or local injection of MSC, MSC-EV, or MSC-CM has been demonstrated to enhance tubular repair, ameliorate inflammation, alleviate fibrosis, and preserve kidney function. In clinical trials of AKI and CKD (Table 3), MSC therapy evinced safety and tolerability, but the small case number and short duration of follow-up meant that the efficacy of MSC therapy on kidney protection was inconclusive [172]. For DKD, MSC administration tended to stabilize or improve the GFR in patients with type 2 DM, but hyperglycemia may diminish the renoprotective effect by causing damage to MSCs, which could be resolved by a co-culture of macrophage with MSCs or a modification of MSCs by angiotensin-converting enzyme 2 [173]. Similarly, uremic toxins such as p-cresol can impair the function of MSCs, and the concomitant use of pioglitazone helps to restore the MSC function [174].

## 9. Conclusions

The management of patients with CKD is complex and challenging because of the heterogeneous nature of CKD and an inadequate awareness of its adverse impact. Besides lifestyle modifications and the correction of CKD risk factors, the prompt identification of AKI and early medical interventions for complications including anemia, metabolic acidosis, low protein diet with ketoanalogues, hyperkalemia, and CKD-mineral bone disorder (CKD-MBD) are crucial for retarding renal progression and lowering the morbidity and mortality of patients with CKD. With the advancements that have been made in understanding the molecular mechanisms of CKD, novel therapeutic agents being continuously identified, and the translation of these findings from bench to bedside will likely improve the efficacy of treatments for CKD.

## Figures and Tables

**Figure 1 ijms-22-10084-f001:**
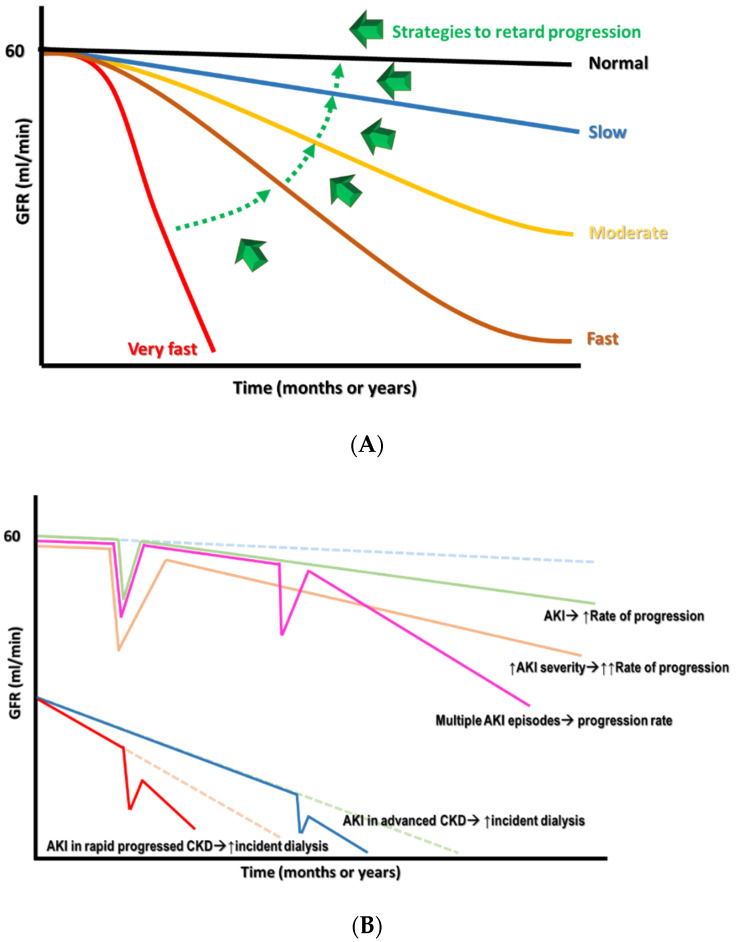
(**A**) The different declined rates of renal function in CKD with the target switch from superfast to slow rate. (**B**) The consequence of AKI on CKD progression, depending on the severity and frequency of episodes.

**Figure 2 ijms-22-10084-f002:**
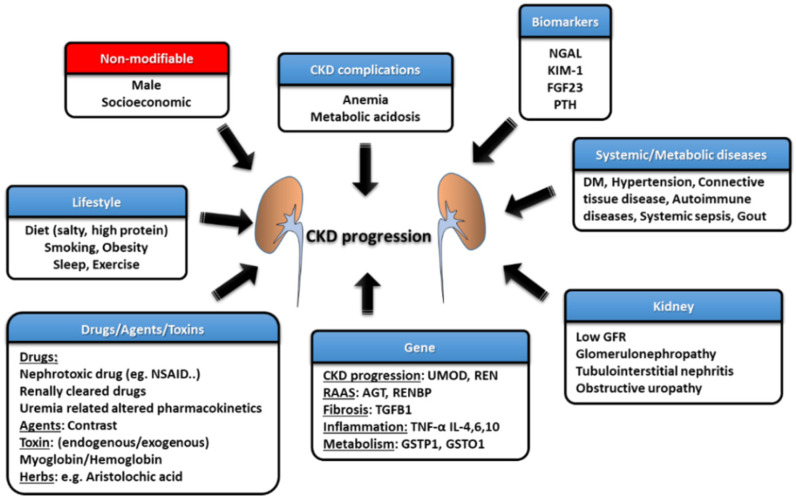
The risk factors and management strategies of CKD development and progression.

**Figure 3 ijms-22-10084-f003:**
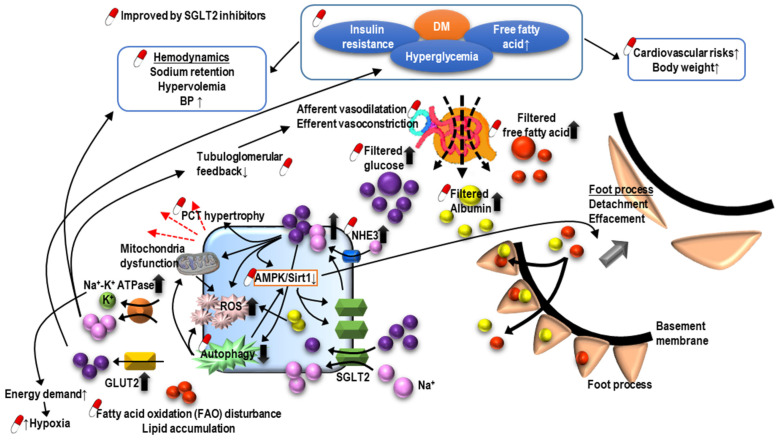
The pathogenesis of diabetic nephropathy and mechanism of sodium-glucose cotransporter inhibitors-associated renoprotection.

**Figure 4 ijms-22-10084-f004:**
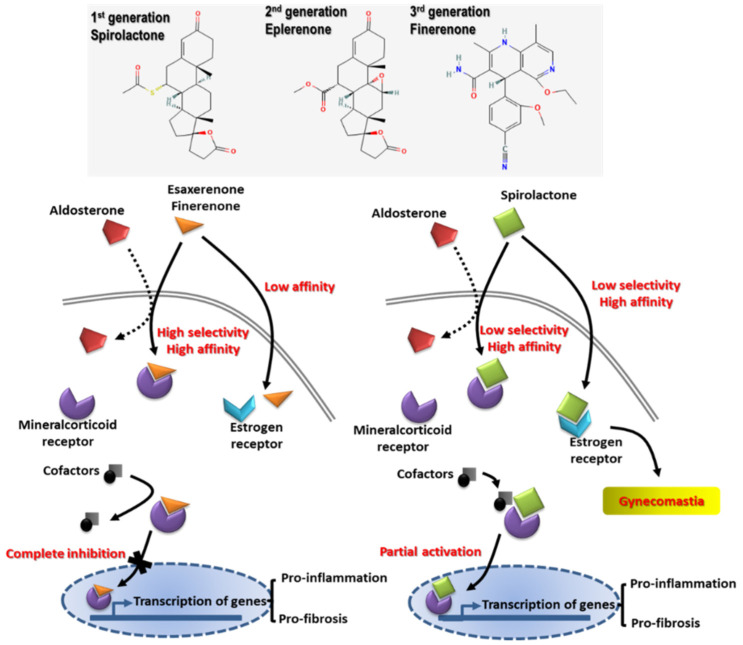
The structure and action mechanisms of different MRAs generations such as spironolactone, eplerenone, and novel nonsteroidal MRAs (esaxerenone and finerenone). Novel nonsteroidal MRAs exert better anti-fibrotic and anti-inflammatory effects with renal tubular sparing effects on hyperkalemia.

**Figure 5 ijms-22-10084-f005:**
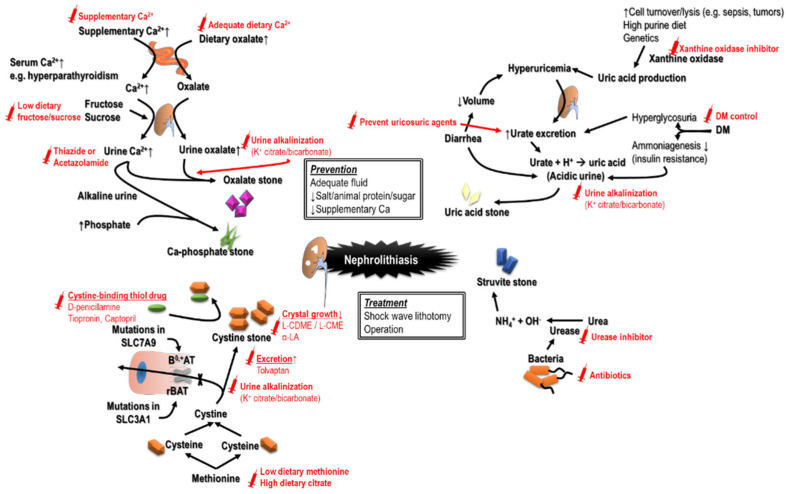
Different causes of nephrolithiasis and urothithiasis with the recommended management strategies to avoid recurrence.

**Figure 6 ijms-22-10084-f006:**
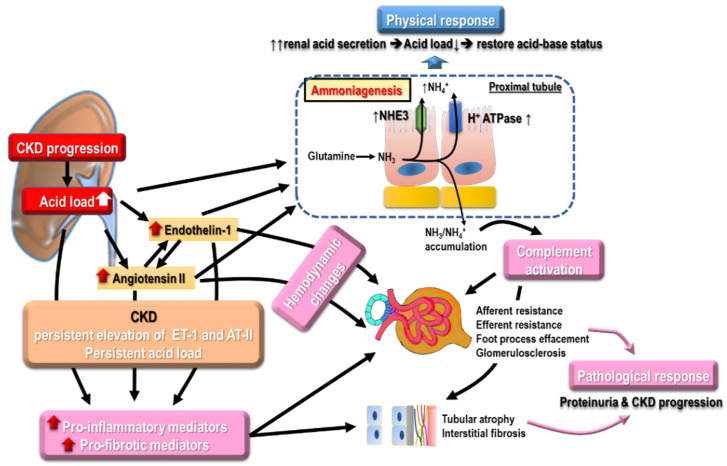
The multiple-faceted mechanisms of metabolic acidosis on renal progression such as enhanced ammonia production in surviving nephrons with complement activation and increased endothelin production, leading to reduced GFR and augmented tubulointerstitial injury.

**Table 1 ijms-22-10084-t001:** Common etiologies of acute kidney injury.

Categories	Mechanism	Examples	Evaluation
Prerenal	Cardiac output↓	Acute myocardial infarction, valve rupture, acute pericarditis, acute myocarditis Drugs exacerbate heart failure (COX inhibitors, CCB, TZD, DPP-4i) Drugs cause direct heart injury (rheumatologic agents (e.g., TNF-α inhibitors), anthracyclines, taxanes, targeted therapy (e.g., bevacizumab, sorafenib), anti-Parkinson (Pergolide, Pramipexole)	**History:** fever, vomiting, diarrhea, chest pain, orthopnea, palpitation, urine output↓, liver/CV diseases Drug: diuretics, NSAID **Physical exam:** BP↓/HR↑, skin turgor/mucosa, edema
	True hypovolemia	Renal loss (diuretics, osmotic diuresis); Extrarenal loss (diarrhea, hemorrhage, burn, third spacing)	
	Effective volume↓	Sepsis, neurogenic shock, anaphylaxis	
	Intrarenal vasoconstriction	Hypercalcemia, hepatorenal syndrome, drugs (CNIs, NSAID, vasoconstrictors.)	
Intrinsic	Glomerular injury	Nephrotic (MCD, MPGN, drugs (NSAID, gold, penicillamine))213607Nephritic (IRGN, lupus nephritis, AAV, anti-GBM disease, IgAN, drugs (e.g., hydralazine))	**History:** Fever, cellulitis, URI, flank pain, foamy urine, urine output↓, myalgia, hemoptysis Drug: antibiotics, NSAID, statin, contrast **Physical exam:** BP, Skin rash, arthritis
	Tubular injury	Severe prerenal causes, toxins (endogenous: hemolysis, rhabdomyolysis, tumor lysis syndrome) or exogenous (aminoglycoside, contrast, CNIs, acyclovir, lithium, vancomycin))	
	Interstitial injury	Allergy (drug: cephalosporin, penicillin, PPI, NSAID, herbs); Infection (bacteria, fungus, virus, leptospirosis); Autoimmune (Lupus, anti-TBM disease, AAV)	
	Vascular injury	Small caliber (TMA (malignant hypertension, HUS/TTP, DIC), scleroderma renal crisis) Large caliber (renal infarction, renal vein thrombosis)	
Postrenal	Urinary tract	Benign prostatic hyperplasia; neurogenic bladder; Intra-ureter (stones, tumors); Extraureter (retroperitoneal fibrosis, intra-abdominal tumors) lesions	**History:** low urinary tract symptoms, gross hematuria **Physical exam:** suprapubic tenderness, abdomen mass **Image:** e.g., ultrasound
	Intrarenal	Crystals (acyclovir, indinavir), stones, tumors, paraproteins (myeloma)	

Denote: AAV: ANCA associated vasculitis, COX: cyclooxygenase, CNIs: calcineurin inhibitors, DPP-4i: Dipeptidyl peptidase 4 inhibitors, DIC: disseminated intravascular coagulation, HUS: hemolytic uremic syndrome, IgAN: IgA nephropathy, MCD: minimal change disease, MPGN: membranoproliferative glomerulonephritis, NSAID: nonsteroidal anti-inflammatory drug, PPI: proton pump inhibitor, TBM: tubule basement membrane, TMA:thrombotic micrangiopathy, TNF: tumor necrosis factor, TTP: thrombotic thrombocytopenic purpura, TZD: Thiazolidinedione.

**Table 2 ijms-22-10084-t002:** The therapy of glomerulopathy and the common side effects of the pharmacotherapy.

	Non-Immune Therapy		Immunosuppressant Therapy						Denote
	↓Dietary salt/protein SGLT2 inhibitors RAAS blockades Blood pressure Infection prophylaxis (vaccine, antibioticsantiviral agents) Vitamin D + calcium	Diuretics ↑Oncotic pressure (albumin infusion) Lipid lowering Anticoagulation (prevent or treat thromboembolism)	Steroids	Calcineurin inhibitors	Antimetabolite	Alkalizing agents	Anti-CD20	PP	
			Prednisolone ACTH	Cyclosporine Tacrolimus	Mycophenolate Azathioprine	CYC Chlorambucil	Rituximab		
MCD	ν	ν	ν	ν	ν	ν	ν		
FSGS	ν	ν	ν	ν		ν			Genetic test: Congenital/infantile type, *APOL1* (adult)
MN	ν	ν	ν	ν	ν	ν	ν		Serum anti-PLA2R: diagnosis, follow-up and outcomes
MPGN	ν	ν					ν	ν	Treat underlying diseases (e.g., MM, lymphoma or HCV)
IgAN	ν	ν (IgAN+MCD)	ν		ν (some RTCs)				Adjuvant antimalarial Ongoing trial: Fostamatinib, Atacicept, Bortezomib
LN	ν	ν (class V)	ν		ν	ν	ν		Antimalarial agents AZA for maintenance
AAV	ν		ν			ν	ν	ν severe AKI PH	Disease activity: chemokine C-X-C motif chemokine ligand 13, matrix metalloproteinase-3, tissue inhibitor of metalloproteinases-1
Ani-GBM	ν		ν			ν		ν till anti-GBM (-)	Overlap syndrome (ANA, ANCA)
Common S/E	Rare	Rare	↑Glucose Cushing ↑BP	Nephrotoxic	GI upset Leukopenia	Bone marrow suppression Infertility	Infusion reaction, Infection Cytopenia	Fever Urticaria	

Denote: AAV: ANCA associated vasculitis, BP: blood pressure, CYC: cyclophosphamide, FSGS: focal segmental glomerulonephropathy, GI: gastrointestinal, GBM: glomerular basement membrane, IgAN: IgA nephropathy, LN: lupus nephritis, MCD: minimal change disease, MM: multiple myeloma, MN: membranous nephropathy, MPGN: membranoproliferative glomerulonephritis, PH: pulmonary hemorrhage, PP: plasmapheresis, RAAS: renin-agiotensin-aldosterone system, SGLT2: sodium glucose cotransporter 2, S/E: side effect.

**Table 3 ijms-22-10084-t003:** Potential novel agents targeting CKD progression.

Categories	Agents	Mechanism
Anti-fibrotic	Pirfenidone	TGF-β inhibitor
	Fresolimumab	Anti-TGF-β monoclonal antibody
	FG3019	Anti-CTGF monoclonal antibody
Anti-oxidative stress	Bardoxolone methyl	Activating Nrf-2 and inhibiting NF-κB pathway
	Probucol	Phenolic antioxidant
Anti-inflammatory	Adalimumab	Anti-TNF-α monoclonal antibody
	Infliximab	Anti-TNF-α monoclonal antibody
	Etanercept	soluble recombinant receptor targeting TNF-α
	Rilonacept	Anti-IL-1 monoclonal antibody
Signal transduction pathway	Ruboxistaurin	PKCβ inhibitor
	Selonsertib	ASK1 inhibitor
	Fasudil	Rho kinase inhibitor
Vasoactive agents	Avosentan	ET type A receptor antagonist
	Atrasentan	ET type A receptor antagonist
	Sitaxsentan	ET type A receptor antagonist
	Bosentan	Dual ET type A and B antagonist
	Sparsentan	Selective dual antagonist of AT1 receptor and the ET type A receptor.
Phosphodiesterase inhibitor	PF-00489791	Phosphodiesterase V inhibitor
Cell-cell/cell-matrix interaction	GCS-100	Galectin-3 inhibitor

AT1: angiotensin II type I, CTGF: connective tissue growth factor, ET: Endothelin, IL-1: interleukin-1, TGF-β: transforming growth factor-β, TNF-α: tumor necrosis factor-α.

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
