# Peer review of "Chronic Kidney Disease: Strategies to Retard Progression"

_ijms, 2021, doi:10.3390/ijms221810084_

Round 1
Reviewer 1 Report
In this review, Yan and colleagues summarize the current knowledge with regard to interventions retarding CKD progression. This study is timely and relevant. However, I recommend to include the following aspects to get a full picture:
Major:
- Please add a paragraph discussing the current knowledge about tubular epithelial cell injury and maladaptive repair in the transition between AKI to CKD. In addition, the relevant literature should be cited accordingly (Yang Nature Medicine 2010).
- Please add a paragraph discussing the contribution of EMT program to kidney injury and CKD, with the relevant literature referenced (Zeisberg Nature Medicine 2003, Lovisa Nature Medicine 2015, Grande Nature Medicine 2015). In this context, induction of BMP signaling is also a promising therapeutical approach (Zeisberg Nature Medicine 2003, Tampe JCI 2018).
- Epigenetic mechanisms should also be included with regard to AKI and CKD progression. Particularly, DNA methylation have been extensively studied in this context (Bechtel Nature Medicine 2010, Tampe JASN 2014). In addition, DNA methylation as therapeutical target is attractive and should also be discussed (Tampe EBioMedicine 2015, Tampe Kidney International 2017, Larkin PlosOne 2021).
- Recent advances to identify the origin of myofibroblasts and potential therapeutical terabits should also be discussed (LeBleu Nature Medicine 2013, Kuppe Nature 2021).
Minor:
- Please discuss SGLT2 inhibitors and current molecular knowledge in attenuation of CKD progression in more detail since these drugs are currently emerging.
- According to current recommendations, please avoid the term "renal" and use "kidney" instead (e.g. ESKD instead of ESRD).
Author Response
Please see the attachment. Thank you for all the critical comments.

Reviewer 2 Report
This review, entitled “Chronic Kidney Disease: Strategies to Retard Progression”, collects the best knowledge about the latest strategies aiming to slow down the deterioration of renal function and to treat relevant complications in patients with CKD. The scientific collect is very interesting, however, some problems, as indicated below, should be addressed before the document can be considered for publication. This version of the manuscript is not enough complete.
Here, I present all my objections in details.
Minor revision:
-English language and style are fine, moderate spell check is required to ensure that an international audience can clearly understand your text.
-In the text, there are some points in which it is possible to note typing errors. In general, I suggest to review the style of the manuscript according to the guidelines of the journal.
-I suggest to divide the sections into paragraphs and sub-paragraphs, and to add more recent references.
-In figure 1, authors reported the different declined rates of renal function in CKD with the target switch from superfast to slow rate, using mL/min/year. However, the following sentence “mL/min/month” is also reported in the full text, before Figure 1 A. I suggests to authors to explain it better.
-mL or ml? Uniform abbreviation in text and figures.
-In figure 2, I suggest to modify the style of the management strategies of CKD
-Af is atrial fibrillation. Modify in AF
-Polymorphisms in genes involving the following pathways such as those associated with inflammatory reactions (e.g., tumor necrosis factor (TNF)- α, interleukin (IL)-4), fibrosis (e.g. TGFB1), phase-II metabolism (e.g. GSTP1, GSTO1), CKD worsening (e.g. UMOD) and renin-angiotensin-aldosterone system (e.g. AGT, RENBP) have been suggested to affect the progression rate of patients with CKD. No italics.
-Modify the style of the Table 1 and 2, the colors white and grey have a specific meaning? Better organize columns and rows. In addition, the caption of figure 1 needs to be changed. Table or Table? Uniform the style text. What is the meaning of v in the table 2?
-Better explain the role of mTOR inhibitors.
-Enhanced ammonia production in surviving nephrons secondary to metabolic acidosis in surviving nephrons may result in complement activation with tubulointerstitial damages. Improve this sentence.
Major revision:
-I suggest to improve the section related to the link Diabetes and CKD, adding this recent reference (DOI: 10.3389/fcell.2020.607080), in which the link between O-GlcNAcylation and kidney failure is demonstrated. It is widely established that poorly controlled or uncontrolled diabetes leads to hyperglycemia, and consequently, to increased O-GlcNAcylation in various tissue. O-Glycosylation is the post-translational conjugation of a single monosaccharide, N-acetylglucosamine, to Serine or Threonine residues of nuclear and cytoplasmic proteins, and is involved in many diseases, including diabetic nephropathy.
In addition, the results of this study underscore the essential role of O-GlcNAc modification in governing basic cell homeostatic functions by controlling protein–protein interactions, and further suggest that reducing ICln O-GlcNAcylation may represent a novel strategy in the prevention or treatment of diseases where an RVD derangement might be involved, including chronic complications of diabetes mellitus such as diabetic nephropathy.
- In CKD patient, there is a close association between nutrition and physical activity (DOI: 10.1159/000355784). The authors could also analyse this latter aspect.
Author Response

(The authors gave the same response as above.)

Round 2
Reviewer 1 Report
Thank you for modifying as suggested, I recommend acceptance.
Minor:
In the section about epigenetic mechanisms, it should be written "RASAL1".
Reviewer 2 Report
Accept in present form